# The mental health prognosis of offspring born of genocidal rape is influenced by family members, the community and their perceptions toward them

Fortunée Nyirandamutsa[1] *, Japhet Niyonsenga[1,2], Gaju Kethina Lisette[3], Josias Izabayo[4] *, Emilienne Kambibi[5], Samuel Munderere[5], Célestin Sebuhoro[1], Assoumpta Muhayisa[1], Sezibera Vincent[4]

1 Department of Clinical Psychology, College of Medicine and Health Sciences, University of Rwanda, Kigali, Rwanda, 2 Mental Health & Behaviour Research Group, College of Medicine and Health Sciences, University of Rwanda, Kigali, Rwanda, 3 Global Mental Health MSc Program, King's College London and London School of Hygiene and Tropical Medicine, London, United Kingdom, 4 Centre for Mental Health, College of Medicine and Health Sciences, University of Rwanda, Kigali, Rwanda, 5 Survivors Fund Rwanda, Kigali, Rwanda

* nyirandamutsaf@gmail.com (FN); izabayojosias@gmail.com (JI)

**Data Availability Statement:** All relevant data are within the manuscript and its Supporting information files. Data can also be found on

## Abstract

### Background

There is little known about the family and community maltreatment of the offspring born of the genocidal rape and the offspring's self-perceptions and how they influence their recovery from mental health problems. This study aimed to examine how the mental health prognosis of these offspring could be influenced by the family or community perceptions and attitudes toward them and their self-perception and coping strategies.

### Methods

Thirty-two semi-structured qualitative interviews were conducted on 16 dyads of mothers and their offspring who were selected from countrywide. The interviews were audio-recorded and transcribed verbatims that were analysed inductively using thematic analysis within the NVivo 12 software.

### Results

Participants reported long-term psychological and psychosomatic consequences stemming from being born of genocidal rape. Notably, family and community maltreatment of the offspring and their self-perception exacerbated psychological distress and affected their capacity to recover. The majority of the offspring were using coping strategies such as sole collaboration with peers with the same history, efforts to hide their birth history, social Isolation (silence, untrusting, involvement in media etc), hardworking, reversed roles in the parental relationship, extreme involvement in praying, and harmful alcohol use.

Zenodo via the following URL: https://zenodo.org/records/10495470.

**Funding:** The author(s) received no specific funding for this work.

**Competing interests:** The authors have declared that no competing interests exist.

**Abbreviations:** BR, born of rape; FARG, Genocide Survivors Assistance Fund; IRB, The institutional review board; RM, raped mother; SURF, Survivors Rwanda Foundation; TVET, Technical and Vocational Education and Training Programme; UK, United Kingdom.

## Conclusion

Given the documented detrimental effects of individual, family and community attitudes and perceptions on psychological, and psychosomatic symptoms as well as the offspring coping strategies, culturally relevant mental health interventions are required to support the well-being and social reintegration of individuals born of genocidal rape while minimizing stigma and their maladaptive coping strategies.

## 1. Introduction

Rape was a major issue during the 1994 genocide against the Tutsi, and its results had long-term severe effects on the mental health and psychosocial living conditions of the victims and the offspring born of this sexual violence [1]. During the Rwandan genocide, men, predominantly Hutu, utilized the rape of women, primarily Tutsi, as a political weapon to exterminate the Tutsi ethnic group [2]. Between April 6 and July 12, 1994, Tutsi girls and women were raped throughout Rwanda especially after mid-May when Hutu officials directed the Interahamwe ("those who stand together") militia not to spare Tutsi women and children in the genocide [2]. They utilized genocidal rape to humiliate, kill and eradicate the Tutsi as an ethnic group [3,4]. Though the exact number of children born as a result of genocide rape is unknown, it is believed that between 2,000 and 10,000 offspring [5,6], were born from 250,000 to 500,000 estimated rapes of Tutsi women and girls [6,7]. Many Tutsi women were raped multiple times and gang-raped; thus, many children born due to the rape do not know their fathers or the families from which they came.

However, mothers find it difficult to tell their rape-born children how they were born since it would require a lot of energy from both mothers and children. Therefore, mothers desire to keep it secret, but it is difficult because the children have the right to know their dads [8]. These offspring are deeply concerned by the societal violent abuse that resulted from their birth as well as their treatment by society. They may be particularly susceptible and experience difficulty in their daily lives as a result of their tremendous difficulties in life and lack of stable familial networks [9]. They may have challenges, including health issues, as a result of their birth conditions, and their mothers' psycho-social trauma may harm their early childhood and adulthood development [9].

Therefore, mothers are the primary victims of genocide rape but their children are also victims [10,11]. While their birth is not a crime against the children, they become victims as a result of the crimes perpetrated against their mothers [12]. These children are vulnerable to hazards to their health and well-being from conception, with the possibility that these threats will persist throughout their lives. Thus, as the children born of genocide rape grow into adulthood, we find it imperative to explore how the perceptions, beliefs and attitudes of the family and community toward the offspring and their self-perception are still affecting their mental health in adulthood, and hence reduced capacity to recover from mental health problems. For example, because they are identified as the perpetrator's children, they are more likely to be abused as children [4,6] which may affect both self-perception and recovery. Culturally, a child belongs to his father, not his mother [13]. Thus, Tutsi survivors refer to these children as Interahamwe, whilst Hutu rapists' relatives frequently abuse the children's mothers for testifying against their dads and sending them to jail.

Worryingly, the majority of what we know about these children and their lives comes from their mom's perspectives [6,14]. As the children born of genocide rape grow into early

adulthood, it is critical to understand how their early life experiences and community perceptions affect their current health and well-being. A small number of published research documents children's experiences and needs from their viewpoints. Furthermore, existing studies are overly broad in this sense. For example, coping with children born from rape in Rwanda [15], parenting style and its psychological impact [16], and the Importance of understanding sexual violence in conflict [16]. This study aims to explore how the family and community attitudes and perceptions of the children conceived via genocidal rape and the offspring's self-perception would affect the recovery from mental health problems. This study also sought to reveal how the offspring are coping with these perceptions and mental health problems.

By understanding experiences and coping strategies (i.e. maladaptive) from the children's perspectives, we can work toward designing effective interventions to prevent or mitigate the potential consequences of adverse life experiences related to being conceived through genocidal rape. This research can also help us better understand how we can break the cycle of intergenerational trauma by supporting the youth born of genocidal in their future parenting and parent-child relationships as well as learning adaptive coping strategies. The studies focusing on the psychological problems of these children and the social problems they face must be expanded.

## 2. Method

### 2.1 Study design

A qualitative study design was conducted to explore how the family and community attitudes and perceptions of the offspring born of genocidal rape and the offspring's self-perception would affect the recovery from mental health problems.

### 2.2. Research approval

The department of Clinical Psychology at the University of Rwanda and the Survivors Rwanda Foundation (SURF), provided initial research approval. The SURF was enormously involved in the study to ensure that all materials used were suitable. The institutional review board (IRB)of the College of Medicine and Health Sciences at the University of Rwanda provided ethical approval (No 174/CMHS IRB/2021).

### 2.3. Participants

Participants of the current study were 16 mothers raped in the 1994 Genocide against the Tutsi and their 16 offspring born of this rape who were selected from countrywide. This sample size was determined by data saturation, a stage in the study process when data analysis reveals no new information, and this redundancy alerts researchers that data collecting may end at this moment [17]. The respondents were approached through the Survivors Rwanda Foundation (SURF), a non-governmental organization that assists genocide survivors throughout the country. SURF is now giving education and counselling assistance to children conceived from the 1994 genocide against the Tutsi genocide. FARG (Genocide Survivors Assistance Fund), a government agency that assists vulnerable genocide survivors, poses a problem for these raped mothers and their offspring born of genocidal rape because it does not consider these children eligible for support because they were born after the genocide and are thus not survivors by definition [18]. As a result, SURF determined that it was critical to assist this specific population.

Supported by SURF, a purposive sample of 32 participants (16 dyads of mother and their offspring) participated in the current study. The inclusion criteria were women of any age (i)

who had a last-born offspring conceived from genocidal rape and ii) who has not been married since the rape. On the offspring, the inclusion criteria were (i) being born of rape, (ii) having lived with the mother from birth to 18 years and (iii) being single. Both mothers and their offspring were excluded from the study if they had (i) severe physical and mental illnesses that would affect their judgement or (i) were married. Additionally, unmarried women who were raped and regularly gave birth were excluded from this study to harmonize the context since they were shown to face significant social stigma in their community [19].

## 2.4. Data collection

The data was collected from December 13, 2021, to April 2, 2022. All participants (mothers and children) were invited to the SURF office to participate in this study, facilitated by the SURF counsellor who works with these individuals at SURF. The participants were thoroughly informed about the research aims and methodology by the primary investigator and the SURF counsellor. Participants gave their verbal and written consent. They were advised of their ability to withdraw from the research if they did not wish to participate or if they had changed their minds. Participants had access to psychosocial support networks since the interviews might lead them to re-experience psychological discomfort. The research team, which included a local psychologist, did post-interview follow-ups to check if the participants were doing well.

Individual interviews in Kinyarwanda were performed in secure and comfortable settings (counselling room of SURF office) and audio-recorded with authorization. The interviews were carried out separately for children and mothers. The interviews lasted 30 and 40 minutes. Open-ended questions like "Please tell me your life story, and share with me whatever you believe is significant" were used to start interviews. The researcher gave the respondents the freedom to discuss any subjects they wanted and, in any sequence, they desired. The following questions were posed during the interviews:

1. Could you tell us about the image people attribute to you (family and community members)?

2. Could you share with us what you think about yourself, and your perception related to your life story?

3. Could you discuss how the image family or community members attribute to you has affected you (psychologically, socially etc.)?

4. Could you describe the coping strategies you use to manage your situation?

**Ethical approval and consent to participate.** Ethical approval was obtained from the Institutional Review Board of the University of Rwanda, CMHS (IRB-CMHS). Both verbal and written informed consent were obtained from the participants after explaining clearly the research objectives and research process. In addition, participants were informed that no benefits or risks were associated with their participation in the study. Finally, the Declaration of Helsinki was respected. This states that participants have the right to opt-out of the study if they don't want to participate or if they change mind.

## 2.5 Data analysis

The framework for thematic analysis in psychological research was used within the NVivo 12 software [20]. The goal of thematic analysis is to find reoccurring themes and patterns of significance in data [20,21]. The analysis was divided into six steps. To improve traceability, a

brief explanation of each process is supplied [22]. The first step involved analyzing audio recordings of the interviews, which was followed by a detailed transcription that was cross-checked against the tapes and by a second researcher. Step 2 entailed identifying fundamental elements of the full data set that were pertinent to the research questions [20]. Certain terms and syntax from the transcriptions were used to guarantee that raw data and codes were in close proximity [23].

The NVivo software (version 12) was then used to upload the transcripts and continue the process of developing the first codes from raw data using an inductive technique, as well as to discuss the original codebook. Two different scholars coded the transcriptions individually. Two more researchers then compared and reviewed the initial codes, which was necessary to minimize the individual impacts on coding [24]. The list of detected codes was evaluated in step 3 in attempt to restructure codes into possible themes. Quotes from the transcriptions were also included. Possible themes and subthemes were developed, contrasted, and debated. Each theme and subtheme were allocated a colour code to keep track of the evolving topics.

In step 4, themes and subthemes were revised and checked against the coded data to ensure that they were based on participant information. A check against the full data set assures inner and outward homogeneity [25]. Codes grouped by topic were compared to other codes and themes. This resulted to certain codes being restructured and subsequently placed in a new theme. In phase 5, there was a final examination of themes and their potential importance for the study and the research question. The 'essence' or 'core' of each theme and subtheme was determined and listed. Subthemes were introduced if they were thought to be important to the theme structure or complexity. Step 6 entailed presenting each topic in a report, with an emphasis on each theme story from phase 5. To substantiate conversations and arguments, quotes from all participants were incorporated. The same four researchers carried out all phases. Notably, the interviews were recorded and analysed in Kinyarwanda before being translated into English. This decision was taken because meaning would be readily lost in translation and the data's authenticity would be jeopardized.

## 3. Results

### 3.1 Sociodemographic characteristics

This study included 16 mother-child dyads as participants. The respondents were drawn from ten districts, including many from Kamonyi (8/32), Nyarugenge (7/32) and Gisagara (n = 5/32). At the time of the study, all mothers were single (8/16) or windowed (8/16). In terms of education, many mothers did not finish primary school (9/16), followed by TVET (5/16), but many children completed university (7/16), followed by TVET (5/16), and advanced level (3/16). Half of the mothers (8/16) and all of the offspring (16/16) were single (Table 1). It was found that 9 out of 16 mothers had only one kid, indicating that they had been raped before marriage. And seven mothers had additional children, implying that they were raped after murdering their husbands.

### 3.2. The influence of family members, the community and own perceptions on the mental health of offspring born of genocidal rape

The analysis of the interviews revealed five key themes related to participants' perspectives on the influence of family or community perceptions and self-perception on recovery from mental health problems. These included: (1) family and community attitudes and perceptions; (2) self-perception and unique challenges; (3) mental disorders symptoms; (4) psychosomatic

**Table 1. Summary of the themes and sub-themes.**

| Themes | Sub-themes | Frequency | |
|---|---|---|---|
| | | Offspring (n = 16) | Mothers (n = 16) |
| **Family and community attitudes and perceptions** | | | |
| | Being stigmatized and rejected by family and community members. | 14 | 11 |
| | Perceived as little Interahamwe and child of a bad memory. | 13 | 9 |
| | Uncontrollable anger and aggression | 6 | 7 |
| | Low-self-confident and incapable individuals | 4 | 3 |
| **Self-perception and unique challenges.** | | | |
| | The effort to take father responsibility | 10 | 9 |
| | Being financially strained | 10 | 8 |
| | Guilt and shame of being the child of Interahamwe | 8 | 5 |
| | Self-stigma | 6 | 5 |
| | Lack of identity or life meaning | 3 | 3 |
| | Bad memories of their mothers | 2 | 3 |
| **Psychological disorders' symptoms** | | | |
| | Avoidance of reminders of their history (conversations, place of origin etc) | 16 | 13 |
| | Depressed mood | 12 | 11 |
| | suicidal ideation and behaviours | 10 | 7 |
| | Social Isolation | 9 | 8 |
| | Sleep disturbance (i.e., insomnia & nightmares) | 9 | 6 |
| | Intrusive negative thoughts | 8 | 5 |
| | Frequent anger and revenge tendencies | 5 | 4 |
| | Inability to speak | 4 | 3 |
| **Psychosomatic symptoms** | | | |
| | Persistent headaches | 12 | 8 |
| | Digestive problems or stomachache | 9 | 7 |
| | Eye pain | 4 | 3 |
| **Coping strategies with family or community attitudes and psychological distress** | | | |
| | Collaboration with peers with the same history | 16 | 13 |
| | Social Isolation (involvement in media, silence, and untrusting of anyone) | 12 | 9 |
| | Efforts to hide their birth history | 13 | 8 |
| | Hardworking spirit | 5 | 3 |
| | Reversed roles in the parental relationship | 4 | 3 |
| | Extreme involvement in praying | 3 | 4 |
| | Harmful alcohol use | 3 | 6 |

symptoms, 5) coping strategies with family or community attitudes and psychological distress that emerged with several sub-themes (Table 1).

**3.2.1. Family and community perceptions of offspring conceived from genocidal rape.** This study revealed that both the family and community have bad perceptions and attitudes toward youth conceived from genocidal rape. The respondents reported being perceived and treated as little Interahamwe and children of a bad memory. These offspring reported being accused of the crimes of their fathers who were the rapists and the perpetrators during the 1994 genocide against the Tutsi.

"When people in the neighbourhood see me getting angry as everyone can do, they say that it is typical of Interahamwe children and that it is a sign that I will be a future killer. It's so painful."

**01-BR (born of rape)**.

"My mother's family hates me because my father was one of those who killed their family members. This makes them treat me like a little killer too because I am an offspring of a genocidaire. Even my mother doesn't love me and I can see it in her eyes, her words or her actions towards me. My birth conditions pushed them to consider me as a permanent memory of the terrible moments they lived during the genocide. . ."

**01-BR**.

The offspring also testified about being stigmatized and rejected by both mothers' and fathers' families and the community members. While the mother's relatives and community members accused them of their fathers' crimes such as rape and killing, the father's relatives accused them of their mother's testimonies that led to the incarceration of the rapists. Following the interview, the researcher revealed that a major portion of our community and peers considers offspring born from rape to be in a group of genocide perpetrators. The peers of these children also saw them as minors or future killers, and they are not proud to be their friends or companions.

"It troubles me to live without love from my both families but also to realize that people of my generation are not interested in friendship with me because of my birth conditions. Even if I am innocent, my mother's family stigmatizes me as I am a son of a rapist and killer while my father's family rejects me because my mother is one of those who accused him of having committed the rape and genocide against the Tutsi"

**05-BR**.

"I did everything to have a good relationship with my both family members but in vain! Even if my mother didn't want it, I tried several times to visit my father's family to familiarize myself with them but they reject me because I represent a source of conflict between the two families. Also, people of my age don't understand my situation and some hesitate to have a deep relationship with me since I can't answer certain questions they ask me about my life story. I really don't know how to explain the pain it causes me. . ."

**12-BR**.

"I didn't have the chance to be loved like my mother's other children even if I am the youngest in the family. Neighbours, colleagues and others who knew my story did not hesitate to tell me that I am a result of the rape experienced by my mother during the genocide. It's very frustrating and it's a constant pain that no one else can understand except the one who lives such a life.. . ."

**06-BR**.

"I grew up with my grandmother because my mother had abandoned me right after my birth. With the death of my grandmother, I was taken by other people who didn't take care of me and used to give me hard work compared to my age"

**07-BR**.

On the mothers' side, the majority (10/16) reported these offspring having uncontrollable anger and aggression and perceiving them as low-self-confident and incapable individuals.

"My daughter is a very angry and resentful person who fears nothing nor anybody, even not me as her mother. It's been a long time since I avoid to comment or correcting her when she makes a mistake because I'm afraid of being attacked or beaten by her. . . she often tends to get revenge on people who have hurt her. Even though I never wanted to have a child born of rape, it is very painful that I cannot advise her or have a good relationship with the single child I have.. . ."

**01-RM (raped mother)**.

"You can't imagine how my son gets irritated when something happens in a way he doesn't want! In this case, he begins to exhibit anger which can even push him to be aggressive towards people who contradict him. I think that the bad words thrown at him by different persons created in him a kind of low self-confidence and a lack of trusting people. He even told me many times that he prefers to be with machines or robots rather than working with human beings. . ."

**04-RM**.

**3.2.2 Self-perception and their unique challenges.** The effort to take the father's responsibility, being financially strained, guilt and shame from being the child of Interahamwe in both family and community, self-stigma, bad memories of their mothers, and lack of identity or life meaning. The following are sample verbatims quotes from the respondents:

"Often, I feel guilty for being a child of a rapist and genocidaire because it makes me a permanent bad memory for many people including my mother. Frankly speaking, my life is complicated because I live like someone who comes from nowhere or like a person who has an unclear identity. . ."

**13-BR**.

"Thinking about my birth conditions stigmatizes me and makes me feel guilty because I know that my mother would not have wanted to have a child as she had me. Despite all this, it is me who must take care of everything that should be done by my father because my mother has no possibilities to satisfy our needs. To be honest, all this does not make my life easier, especially because I also have a challenge related to my dirty origin. . ."

**08-BR**.

"I experienced stigma and guilt since I learned about my birth conditions associated with my undesirable identity. Even if I do everything that I can to support my mother financially, I live with remorse because I am a result which reminds her of the worst experience she lived during the genocide. I know I can't change anything about the story of my existence but it makes me very sad every time I think about it. . ."

**14-BR**.

**3.2.3. Psychological problems of the offspring born of the genocide rape.** By hearing from the experience and needs of the respondents, it can be said that the offspring inherited

their mothers' genocide-related suffering. The findings of this study illustrate that individuals born of genocide-rape suffer different psychological problems caused by their origin and exacerbated by family and community maltreatment, and self-perception. Alarmingly, though these children are now adults, our findings show that humiliation, rejection, and marginalization from family and community members added to their severe suffering and hampered their capacity to heal from their psychological distress and pain.

"The hatred that the members of my both families show me makes me suffer a lot. I remember every day how my maternal uncle used to tell me that I am the source of any disaster that happens to them. This pushes me to start putting myself away from them to prevent them to avoid their bad words which hurt me. Every time I think about it, I feel disturbed and start having trouble sleeping. To relieve me of this pain and to forget my life story, I take refuge in alcohol and come home late at night when they are all asleep..."

**01-BR**

"The conflicts between the members of my two families make me suffer physically and mentally. I remember that there was a time when I was very disturbed because my mother did not want me to visit my father's family and even if I tried to sneak her in, my father's family used to chase me because they don't love me. There was a period when I couldn't control my thoughts and actions until my mother's family took me to go for treatment in mental health services. At the moment I am followed in individual or group counselling sessions and I participate in youth camps. . . ."

**12-BR**.

As confirmed by their mothers, the offspring reported PTSD and depression symptoms such as depressed mood, suicidal ideation, social isolation, sudden inability to speak, sleep disturbance (i.e. insomnia and nightmares), intrusive negative thoughts of their conception history, and avoidance tendencies.

"My child's behaviour is difficult to manage because she changes his mood without a valid reason. She abuses alcohol and other substances and she complains of sleeping troubles. Also, she does not like to socialize with others and runs away from having friends or being attached to anyone. Due to the negative thoughts linked to her birth conditions, she attempted to commit suicide one day! I admit that this suicide attempt scared me and made me sad until today.. . ."

**01-RM**.

"I do not like to involve myself in social issues because I prefer to stay alone and to keep silent. To protect myself against questions related to birth history, I look always at how I can avoid people who can ask me about my private life because talking about it reminds me that I am a result of rape experienced by my mother during the genocide. . ."

**10-BR**.

"My son abuses alcohol and often loses his job because of bad behaviour and lack of determination. He never listens to people who give him good advice, not even me. He frequently remains alone, does not like to share ideas with others and rarely participates in other

people's ceremonies. I have tried several ways to see if he can change but in vain and it pains me a lot. . ."

**11-RM**.

As shared by the offspring themselves and their mothers, antisocial tendencies such as uncontrolled anger, aggression and revenge tendencies were evident in this study.

"Although I try to be patient or to control myself, it is difficult for me to forgive someone who hurts me. If someone betrays me, I start to think about how I can ever get revenge. Even if it's about fighting with someone stronger than me I can do it because sometimes I can't control my anger. . ."

**01-BR**.

- "I had the bad luck of having a daughter who consumes too much alcohol. She drinks all kinds of alcohol with anyone without distinction; she comes home when she wants and is always aggressive. Sometimes, she said she didn't want to live anymore and wanted to kill herself! For the moment, I no longer dare to forbid her to misbehave because when I try to talk to her about it, she intimidates me by telling me that she can beat me. . ."

**01-RM**

**3.2.4. Psychosomatic problems of the offspring born of the genocide rape.**   As supported by mothers' interviews, the offspring reported psychosomatic symptoms such as persistent headaches, digestive problems, stomachache, and eye pain.

"I have been suffering from gastric pain for a long time and I have been to several hospitals for treatment but they all told me that I have no disease. I'm always skinny because I don't like to eat and I keep feeling sick despite that medical exam showing no abnormalities. . ."

**01-BR**.

"I have a serious problem with my eyes because I often feel ocular pruritus despite being treated. Additionally, I also have a persistent headache that doesn't respond well to the pills. Now, I do everything to protect myself from the wind or the sun so that the problem does not get worse. . ."

**05-BR**.

**3.2.5 Coping strategies with family or community attitudes and psychological distress.**   Coping strategies used emerged in several subthemes including collaboration with peers with the same history, social Isolation (involving in media, silence, untrusting of anyone), efforts to hide their birth history, hardworking spirit, reversed roles in the parental relationship, extreme involvement in praying and harmful alcohol use.

"Being with my peers with whom I share the same story is the first thing that can make me happy. When I'm with them, I feel free because I know they don't judge me. I can talk without a problem, I can laugh with them, and I can share with them as much as possible. Even my mother tells me that she sees a positive change every time I come to meet or talk with them. . ."

**01-BR**.

"To be honest, only alcohol is the unique way to help me deal with my situation of being a child of a rapist and genocidaire. I know it's not well to drink a lot of alcohol but I can't find any other alternative. Even my mother begged me to stop drinking myself into a stupor like I do but it is very difficult for me"

**11-BR**.

"Being born of the rape committed during the genocide is like living with a burden that tires me every day! Every time I wake up in the morning I start to see how I will keep myself very busy because I do not like to have enough time to think about myself. For this, I invest myself in embroidery and when I finish I go to the Church as prayers allow me to remain calm despite many problems related to my origins. . ."

**03-BR**.

"Besides losing family members and being raped, my mother also lost a leg during the genocide. As her general health status is very vulnerable, I have to be responsible from a young age to take on certain responsibilities that should be done by a father".

**13-BR**.

Moving away from one's place of origin to another to avoid harassment from the neighbourhood was found as robust stigmatisation coping mechanism in this study. The same reflection was made by the respondents:

"People of the village of my origin have many questions about my birth conditions and identity. . .. I was often told that my father is in prison because he is a rapist and genocidaire. Due to their bad attitudes and harsh words towards me, I grew up with a dream to leave this village once adult, reason why I live now far away from them and I am not nostalgic for my native village. . ."

**02-BR**.

"I am traumatized by humiliation and rejection from my maternal family! I was often ashamed and marginalised by their reason. My mother and I left the native village to escape harassment and stigmatisation. Nowadays, our life is better than before, because where we live no one knows our history or our birth conditions . . ."

**03-BR**

## 4. Discussion

Even though sexual violations such as rape and forced pregnancy have been classified as acts of genocide and crimes against humanity under international law, consideration for the children born as a result of these crimes has been largely absent from international human rights and scholarly discourse [26]. In this way, the intergenerational consequences of genocidal rape have been largely overlooked [26, 27], with special attention paid to the impact of family and community attitudes and beliefs, and offspring self-perceptions on their mental health prognosis. It is not clear under what conditions and to what extent these attitudes, beliefs and perceptions affect the offspring's psychological distress and ability to recover from mental health problems.

As revealed in prior studies, the experiences and perspectives shared by young people and their mothers in this study underscored long-term psychological and psychosomatic consequences for offspring born of genocidal rape [4,27]. Our findings also indicated that family and community attitudes and beliefs, and offspring self-perceptions worsened psychological distress [4] and hence affected the capacity to recover. As confirmed by their mothers, the offspring reported post-traumatic stress disorder (PTSD) and depression symptoms such as depressed mood, suicidal ideation, social isolation, sudden inability to speak, sleep disturbance (i.e. insomnia and nightmares), intrusive negative thoughts of their conception history, and avoidance tendencies. Consistently, several scholars have shown that children born as a result of rape are at elevated risk of developing mental health issues such as PTSD depression, anxiety, and antisocial personality symptoms among these offspring [4,28–30].

Notably, our findings highlighted that family and community attitudes and beliefs, and offspring self-perceptions played role crucial role in the development and worsening of these psychological disorders' symptoms. Our findings revealed that these offspring were perceived and treated as little Interahamwe and children of bad memory in family and community. They are stigmatized and rejected by both mothers' and fathers' families and the community members, especially their peers. Due to being perceived and treated as minors or future killers, their peers are not proud of being their friends or companions. In congruence with our findings, several scholars have revealed that young adults born as a result of genocide rape in Rwanda have difficult parent-child interactions, discrimination, stigmatization and rejection, and identity challenges [9,12,31,32]. Also, due to parental psychological problems such as depression, anxiety, or PTSD etc., these offspring may be at heightened risk of being abused and neglected by their mothers [32–34]. As rape-induced pregnancy is viewed as an added traumatic stressor where the offspring is seen as a living reminder of the rape and rapist, the mothers may impose abusive parenting styles on these offspring [12].

Offspring on another hand reported guilt and shame, self-stigma, lack of identity or life meaning and economically strained that may be stemmed from their self-perception as children of rapists and Genocidaire and bad memories of their mothers, and these family and community rejection and stigmatizing attitudes and beliefs [4]. These offspring are severely abused, neglected and stigmatized when the fathers have killed their mother's families and the neighbours in the community [13,28]. Offspring were also found to experience guilt and reversed roles in the parental relationship, becoming carers of their mothers.

This study demonstrated that not only did offspring experience great psychological disorders owing to their identity, but also that their psychosocial conditions were further aggravated by being stigmatized, marginalized, and harassed as a result of conceived from genocide rape. Worryingly, all of these attitudes and beliefs from family and community as well as offspring' self-perception may undoubtful exacerbate psychological disorders symptoms and hence reduced capacity to recover. In the same vein, several scholars have found that family and community rejection [35], and stigma (i.e., public and internalized stigma) have been positively related to poorer psychological and physical health outcomes [9,36]. Stigma is a complicated process that comes from the interaction of the individual with social and cultural norms and involves labelling and stereotyping of the individual [37]. As Rwanda is a strongly patriarchal society, the identity of these offspring is inextricably linked to the rapist father even if he/she has never met or known the father [13].

Of worry, our findings have indicated that the respondents reported using maladaptive coping strategies with psychological distress and stigmatizing identities. They reported sole collaboration with peers with the same history, hiding their birth history, social Isolation (silence, untrusting, involvement in media etc), avoidance of reminders of their history (place of origin, related conversations etc), extreme involvement in praying, and harmful alcohol use among

others, which have detrimental effects on the mental health. Consistently, authors have shown that hiding unresolved trauma and concealing stigmatized identities can be psychologically taxing and socially devaluing [36]. Also, the youth may abuse alcohol and involved in the media as self-medication for their psychologically distressing symptoms [38–40]. In particular, meeting others in a similar situation was found to be a healthy coping strategy as many imagined they were the only ones dealing with this circumstance. This confirms the work of other scholars who states that most victims seek resolution of their traumatic experience by associating with other individuals who have experienced the same event [9,41].

### 4.1 Study limitations

This study has numerous limitations that should caution against generalizing the findings. First, as with any self-report data, information acquired from the participants was impacted by their willingness to share personal information. Because of probable shame and blame, some of the respondents may have been unable to communicate their experiences. Second, the sample may be overrepresented the mothers and their offspring with disadvantaged Rwandan women with poor socioeconomic status as they were beneficiaries of SURF Rwanda. Therefore, there are no viewpoints from individuals from a high socioeconomic class. Finally, the qualitative methodology of the study restricts the generalizability of the findings.

### 5. Conclusion

Overall, the findings that emerged from this study shed light on the mental health problems of the offspring born of sexual violence during the genocide who are now turning 28 years, and the extent the family and community attitudes and beliefs toward these offspring and their self-perception keep aggravating the youth psychological distress. Alarmingly, our study revealed that the youth were adopting maladaptive coping strategies with these psychological distressing and the stigma, harassment, discrimination and rejection from family and community that may have further detrimental effects on their mental health. These study findings call upon culturally relevant mental health interventions to support the well-being and social reintegration of individuals born of genocidal rape while minimizing stigma and their maladaptive coping strategies. Clinicians and professionals that engage with these families must address stigma and discrimination on a community level to ensure that youth born of rape are treated with dignity and helped to lead productive lives. They may easily aid the young people by discussing and creating possibilities for them to develop a more positive self-identity and expand their support networks. They will then have the strength and resources to deal with the unpleasant feelings and help their child during the challenging process of rebuilding a positive identity.

### Supporting information

**S1 File.**
(SAV)

**S2 File.**
(PDF)

### Acknowledgments

We are very grateful to the participants of this study who willingly accepted to participate in the study. We are also grateful to SURF Rwanda for their continual support at the study site.

## Author Contributions

**Conceptualization:** Fortunée Nyirandamutsa, Célestin Sebuhoro, Assoumpta Muhayisa, Sezibera Vincent.

**Data curation:** Fortunée Nyirandamutsa, Japhet Niyonsenga, Gaju Kethina Lisette, Josias Izabayo, Emilienne Kambibi, Samuel Munderere.

**Formal analysis:** Fortunée Nyirandamutsa, Japhet Niyonsenga, Gaju Kethina Lisette, Josias Izabayo.

**Investigation:** Fortunée Nyirandamutsa.

**Methodology:** Fortunée Nyirandamutsa, Japhet Niyonsenga, Gaju Kethina Lisette, Josias Izabayo, Emilienne Kambibi, Samuel Munderere, Assoumpta Muhayisa, Sezibera Vincent.

**Project administration:** Fortunée Nyirandamutsa.

**Resources:** Fortunée Nyirandamutsa.

**Software:** Fortunée Nyirandamutsa.

**Supervision:** Célestin Sebuhoro, Assoumpta Muhayisa, Sezibera Vincent.

**Validation:** Fortunée Nyirandamutsa, Japhet Niyonsenga.

**Visualization:** Fortunée Nyirandamutsa, Japhet Niyonsenga, Josias Izabayo.

**Writing – original draft:** Fortunée Nyirandamutsa, Japhet Niyonsenga, Gaju Kethina Lisette.

**Writing – review & editing:** Fortunée Nyirandamutsa, Japhet Niyonsenga, Gaju Kethina Lisette, Josias Izabayo.

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
