## [Decision Letter · Decision Letter 0]

6 Dec 2023

PONE-D-23-29965The mental health prognosis of offspring born of genocidal rape is influenced by family members, the community and their perceptions toward them.PLOS ONE

Dear Dr. Izabayo,

Thank you for submitting your manuscript to PLOS ONE. After careful consideration, we feel that the article has merit but it needs few revisions. Therefore, we invite you to submit a revised version of the manuscript that addresses the points raised during the review process. Mainly these observations include: 1. Review of literature is not sufficient and the discussion section does not elaborate the findings in detail. You are requested to add some relevant and recent literature to provide support to the objectives, hypotheses and discussion of the study.2. There are some grammatical and typing mistakes, which needs to be reviewed.3. Provide rationale for why the participants include mothers only?

We look forward to receiving your revised manuscript.

Kind regards,

Shazia Khalid, PhD

Academic Editor

PLOS ONE

4. PLOS requires an ORCID iD for the corresponding author in Editorial Manager on papers submitted after December 6th, 2016. Please ensure that you have an ORCID iD and that it is validated in Editorial Manager. To do this, go to ‘Update my Information’ (in the upper left-hand corner of the main menu), and click on the Fetch/Validate link next to the ORCID field. This will take you to the ORCID site and allow you to create a new iD or authenticate a pre-existing iD in Editorial Manager. Please see the following video for instructions on linking an ORCID iD to your Editorial Manager account: " ext-link-type="uri" xlink:type="simple">https://www.youtube.com/watch?v=_xcclfuvtxQ".

5. Please include the reference section of your manuscript.

Reviewers' comments:

Reviewer's Responses to Questions

**Comments to the Author**

1. Is the manuscript technically sound, and do the data support the conclusions?

Reviewer #1: Yes

Reviewer #2: Yes

2. Has the statistical analysis been performed appropriately and rigorously? 

Reviewer #1: Yes

Reviewer #2: Yes

3. Have the authors made all data underlying the findings in their manuscript fully available?

Reviewer #1: Yes

Reviewer #2: No

4. Is the manuscript presented in an intelligible fashion and written in standard English?

Reviewer #1: Yes

Reviewer #2: Yes

5. Review Comments to the Author

Reviewer #1: Overall the research paper impressively delves into a crucial topic but I would like to share few recommendations.

1. Paper should include a more robust theoretical background. Strengthening this aspect would provide a deeper foundation for the study contributing to a more comprehensive understanding of the study variables.

2. The paper's quality is elevated by the careful attention given to its methodology; however, including the age range of the participants in the Method section would add valuable detail and enhance study's applicability.

3. According to my understanding, Line no. 200 ( Half of the mothers (8/16) is contradicting with the inclusion criteria described in Line 132 (were married). If others are widow then mention this also in inclusion criteria.

4. Line 121-124 requires rephrasing as it is not giving a comprehensive understanding of the subject matter.

please note some typing mistakes.

Line 197: windowed should be widow

Line 132: (i) should be (ii)

Reviewer #2: The research titled "The Influence of Family and Community Perceptions on the Mental Health Prognosis of Offspring Born of Genocidal Rape" makes a noteworthy contribution to the existing body of knowledge pertaining to the mental health of individuals born of genocidal circumstances. Its distinctive perspective, methodological rigor, and actionable recommendations contribute to its significance. However, to enhance the study's effectiveness and applicability, certain areas for improvement have been identified.

Primarily, the study's title implies a focus on offspring born of genocidal rape, yet the inclusion of mothers in the sample requires clarification of the rationale behind this decision. Explicitly addressing the purpose and relevance of involving mothers would contribute to the study's transparency and coherence.

There is also dearth of theoretical explanation of finding. By taking into account the theories may help to further explanation of the factors.

While the research highlights the necessity for culturally relevant mental health interventions, it falls short in providing an in-depth exploration of the specific cultural nuances influencing the mental health of the studied population. A more thorough discussion of cultural factors is imperative to inform the development of interventions that resonate with the cultural context.

Furthermore, the study lacks an exploration of external factors that could potentially contribute to the mental health outcomes of the offspring. Factors such as socioeconomic status, access to mental health resources, and ongoing socio-political conditions are integral components influencing mental health. A more comprehensive discussion of these external factors would fortify the study's contextual understanding and relevance.

The research significantly contributes to the field, addressing these identified areas for improvement would strengthen its overall impact and align it more closely with best practices in academic research.

6. PLOS authors have the option to publish the peer review history of their article (what does this mean?). If published, this will include your full peer review and any attached files.

Reviewer #1: No

Reviewer #2: No

---

## [Author Response · Author response to Decision Letter 0]

17 Mar 2024

All comments were thoroughly addressed in the attached "Response to Reviewers" file.

---

## [Editor Report · Decision Letter 1]

2 Apr 2024

The mental health prognosis of offspring born of genocidal rape is influenced by family members, the community and their perceptions toward them.

PONE-D-23-29965R1

Dear Author,

We’re pleased to inform you that your manuscript has been judged scientifically suitable for publication and will be formally accepted for publication once it meets all outstanding technical requirements.

Kind regards,

Shazia Khalid, PhD

Academic Editor

PLOS ONE